# Prevalence and associated factors of treatment failure among children on ART in Ethiopia: A systematic review and meta-analysis

**Belete Gelaw**[1]*, **Lemma Dessalegn**[1], **Eyasu Alem**[1], **Tiwabwork Tekalign**[2],
**Tadele Lankirew**[2], **Kirubel Eshetu**[2], **Chalie Marew**[3], **Bogale Chekole**[4], **Amare Kassaw**[3]

1 Department of Pediatric and Child Health Nursing, School of Nursing, College of Health Science and Medicine, Wolaita Sodo University, Wolaita Sodo, Ethiopia, 2 Department of Nursing, School of Nursing, College of Health Science and Medicine, Wolaita Sodo University, Wolaita Sodo, Ethiopia, 3 Department of Pediatric and Child Health Nursing, College of Medicine and Health Sciences, Debre Tabor University, Debre Tabor, Ethiopia, 4 Department of Pediatric and Child Health Nursing, College of Medicine and Health Sciences, Wolkite University, Wolkite, Ethiopia

* beletegz12@gmail.com

**Data Availability Statement:** All relevant data are within the paper and its Supporting information files.

## Abstract

### Background

As the use of antiretroviral therapy (ART) increases, the issue of treatment failure is still a global challenge, particularly in a resource limited settings including Ethiopia. The results of former studies in Ethiopia were highly variable and inconsistent across studies. Thus, this systematic review and meta-analysis intended to provide the pooled estimation of treatment failure and associated factors among children on antiretroviral therapy.

### Methods

We searched international databases (i.e., PubMed, Google Scholar, Web of Science, Ethiopian Universities' online repository library, Scopus, and the Cochrane Library) during the period of February 30 to April 7, 2021. All identified observational studies reporting the proportion of treatment failure among HIV positive children in Ethiopia were included. Heterogeneity of the studies was checked using $I^2$ test and Cochrane Q test statistics. We run Begg's regression test to assess publication bias. A random-effects meta-analysis model was performed to estimate the pooled prevalence of treatment failure.

### Results

The estimated pooled prevalence of treatment failure among children in Ethiopia was 12.34 (95%CI: 8.59, 16.10). Subgroup analysis of this review showed that the highest prevalence was observed in Addis Ababa (15.92%), followed by Oromia region (14.47%). Poor ART adherence (AOR = 2.53, CI: 2.03, 4.97), advanced WHO clinical staging (AOR = 1.66, CI: 1.24, 3.21), and opportunistic infections (AOR = 2.64 CI: 2.19, 4.31 were found to be significantly associated factors with childhood treatment failure.

**Funding:** The authors received no specific funding for this work.

**Competing interests:** The authors declared that they have no competing interests.

**Abbreviations: AIDS**, Acquired Immune Deficiency Syndrome; **AOR**, Adjusted odd ratio; **ART**, Antiretroviral Therapy; **CI**, Confidence Interval; **COR**, Crud odd ratio; **HIV**, Human immunodeficiency Virus; **TF**, Treatment Failure; **UNAIDS**, Joint United Nations Programme on HIV/AIDS; **WHO**, World Health Organization.

## Conclusions

This study revealed that treatment failure among children on ART was high in Ethiopia. Poor ART adherence, advanced WHO clinical staging, opportunistic infections, and low level of CD4 cell counts increased the risk of treatment failure.

## Introduction

Acquired immune deficiency syndrome (AIDS) is a viral disease caused by human immunodeficiency virus (HIV) that weakens the immune system and makes the body susceptible to opportunistic infections [1]. The Human Immunodeficiency Virus (HIV) pandemic affects many parts of the world population [2]. In 2018, approximately 37.9 million people were living with HIV worldwide. Of which, around 1.8 million were children (age <15 years) [3]. In Ethiopia, 56,514 children were living with HIV in 2018 [4]. In this year, 23.3 million HIV-positive people were gain access to antiretroviral therapy (ART) globally [5].

Antiretroviral therapy (ART) is crucial to decrease progress of HIV/AIDS by suppressing viral replication which in turn reestablishes the immune function of HIV-infected individuals [6,7]. In addition, ART can minimize the risk of HIV transmission [8,9]. However, maintaining long-term adherence level, and viral load suppression, and prevention of treatment failure (TF) remains a serious challenge for HIV-infected children [10]. Even if ART is not curative medicine, it reduces the risk of HIV associated morbidity and mortality [11]. Treatment failure can be assessed based on World Health Organization (WHO) criteria, as immunological, clinical, virological, or combination of those failure [12].

The current guidelines recommend that, virological failure is more informative to assess treatment failure [13]. The Joint United Nations Program on HIV/AIDS (UNAIDS) and partners launched, 90% of individuals on ART would be virally suppressed at the end of 2030 worldwide [12,14]. Therefore, implementing globally recommended preventive measures and early detection of treatment failure is vital for treatment effectiveness and for achieving a stated strategic treatment plan of 2030 [15,16]. Likewise, the Ethiopian government also implemented various preventive strategies, such as timely initiation of ART, increasing knowledge and attitudes of patients towards ART, carrying out food and nutrition policy, prevention and control of opportunistic infections, and improving level of ART adherence [17].

In Ethiopia, studies were conducted to estimate the magnitude as well as to identify associated factors of treatment failure among HIV-infected children [18–27]. These small and fragmented studies demonstrated that TF among children on ART fluctuated from 3.1% in southern Ethiopia [21] to 22.6% in Addis Ababa [27]. However, the existed discrepancy and variabilities have not yet been investigated.

Additionally, there was no a country wide data which represents a national treatment failure and its associated factors among HIV positive children. Hence, the aim of this systematic review and meta-analysis was to estimate the pooled prevalence and associated factors of treatment failure among children on ART using available studies in Ethiopia. The findings of this systematic review and meta-analysis will highlight the prevalence and associated factors of TF with implications to improve health care workers' interventions, assist decision makers and other concerned stakeholders to design, implement and evaluate interventions to improve level of ART adherence. Furthermore, the findings will enable the country to sustain treatment successes and to hasten the decline of childhood treatment failure in Ethiopia.

## Materials and methods

### Reporting

The result of this systematic review and meta-analyses was prepared and reported using the Preferred Reporting Items for Systematic Reviews and Meta-analyses (PRISMA) guideline [28]. The protocol has been registered in the PROSPERO database with a registration number of CRD-248210.

### Study design, settings and search strategies

We carried out a systematic review and meta-analysis to estimate the pooled prevalence of TF and associated factors among HIV-positive children in Ethiopia which is located in east Africa. To find potentially relevant articles, a comprehensive search was carried out on PubMed, Web of Science, Google Scholar, Scopus, and Cochrane Library databases. To find unpublished relevant literatures to this study, some online repository library centers were searched. Besides, gray literatures were searched from review of reference lists and input of content experts. MeSH (Medical Subject Headings), Boolean operators and all fields within records were used to search in the advanced PubMed search engine. Search terms or phrases used were: children, child, pediatrics, treatment failure, antiretroviral therapy, prevalence, and associated factors.

The advanced PubMed database search strategy was performed using the following key terms (((((((Children(tw) OR child(tw) OR pediatrics(tw))) OR (("child"[MeSH Terms] OR "child"[All Fields] OR "children"[All Fields]) AND tw[All Fields] OR ("child"[MeSH Terms] OR "child"[All Fields]) AND tw[All Fields] OR ("paediatrics"[All Fields] OR "pediatrics"[-MeSH Terms] OR "pediatrics"[All Fields]) AND tw[All Fields]))) AND (((Treatment failure (tw) OR immunological failure(tw) OR clinical failure(tw) OR virological failure(tw))) OR (("treatment failure"[MeSH Terms] OR ("treatment"[All Fields] AND "failure"[All Fields]) OR "treatment failure"[All Fields]) AND tw[All Fields] OR (immunological[All Fields] AND failure[All Fields]) AND tw[All Fields] OR (clinical[All Fields] AND failure[All Fields]) AND tw[All Fields] OR (("virology"[MeSH Terms] OR "virology"[All Fields] OR "virological"[All Fields]) AND failure[All Fields]) AND tw[All Fields]))) AND (((ART(tw) OR antiretroviral therapy(tw) OR ARV(tw) OR antiretroviral(tw) OR HAART(tw))) OR (("art"[MeSH Terms] OR "art"[All Fields]) AND tw[All Fields] OR (("anti-retroviral agents"[All Fields] OR "anti-retroviral agents"[MeSH Terms] OR ("anti-retroviral"[All Fields] AND "agents"[All Fields]) OR "anti-retroviral agents"[All Fields] OR "antiretroviral"[All Fields]) AND ("therapy"[Subheading] OR "therapy"[All Fields] OR "therapeutics"[MeSH Terms] OR "therapeutics"[All Fields])) AND tw[All Fields] OR "arv"[All Fields] AND tw[All Fields] OR ("anti-retroviral agents"[All Fields] OR "anti-retroviral agents"[MeSH Terms] OR ("anti-retroviral"[All Fields] AND "agents"[All Fields]) OR "anti-retroviral agents"[All Fields] OR "antiretroviral"[All Fields]) AND tw[All Fields] OR ("antiretroviral therapy, highly active"[MeSH Terms] OR ("antiretroviral"[All Fields] AND "therapy"[All Fields] AND "highly"[All Fields] AND "active"[All Fields]) OR "highly active antiretroviral therapy"[All Fields] OR "haart"[All Fields]) AND tw[All Fields]))) AND (((Prevalence(tw) OR proportion(tw) OR magnitude(tw) OR risk factors (tw) OR associated factors(tw) OR predictors(tw))) OR (("epidemiology"[Subheading] OR "epidemiology"[All Fields] OR "prevalence"[All Fields] OR "prevalence"[MeSH Terms]) AND tw[All Fields] OR proportion[All Fields] AND tw[All Fields] OR magnitude[All Fields] AND tw[All Fields] OR ("risk factors"[MeSH Terms] OR ("risk"[All Fields] AND "factors"[All Fields]) OR "risk factors"[All Fields]) AND tw[All Fields] OR (associated[All Fields] AND factors[All Fields]) AND tw[All Fields] OR predictors[All Fields] AND tw[All Fields]))) AND ((Ethiopia (tw)) OR (("ethiopia"[MeSH Terms] OR "ethiopia"[All Fields]) AND tw[All Fields])). The

search was done between March 3 and April 27, 2021. All papers published up to April 27, 2021 were included. Endnote X8 software manager was used to cite references and manage the searched literatures.

## Measurement outcome variables

This review has two main outcome variables. The first outcome is HIV/AIDS treatment failure which was defined as immunological, clinical, and virological treatment failure [12]. The second outcome for this study was to identify factors associated with treatment failure. For this outcome, we determined the association between treatment failure and associated factors in the form of the log odds ratio.

## Eligibility criteria

**Inclusion criteria.** **Study area**: Only studies conducted in Ethiopia were included to produce single estimate of common effects.

**Study design:** All observational study designs reporting the prevalence of TF were eligible for this meta-analysis.

**Population:** All HIV-positive children on antiretroviral treatment.

**Language:** Only articles reported in English language were incorporated.

**Publication condition:** Both published and unpublished studies were considered.

**Exclusion criteria.** Conference reports and articles without full text access were excluded. Exclusion of these studies is because of the inability to check the quality of articles in the absence of full text.

## Data selection process

The required data from included articles were extracted by two authors (BG and LD) using a standardized data extraction format, adapted from the Joanna Briggs Institute (JBI). Any disagreements during screening were undertaken through discussion. The reviewer contacted the first author(s) of primary research for additional information. The first author name, follow up years, publication year, region, study area, study design, sample size, response rate, and prevalence with 95% CI were included the data selection form.

## Quality assessment

The qualities of included articles were evaluated by four investigators (BG, LD, EA and TT) using Newcastle-Ottawa Scale quality assessment tool for observational studies [29]. The tool has three sections; in which the first section focuses on the methodological quality of each primary study (i.e., sample size, response rate, and sampling technique) and graded out of five stars. The second section considers the comparability of the study cases or cohorts with a probability of two stars to be gained. The third section concerns on outcomes and statistical analysis of the primary study with a possibility of three star scores. Any discrepancies between the two quality assessors were resolved by repeating the procedures and involving third reviewer before computing the final appraisal scores. Articles with a score of $\geq 6$ for cross-sectional and $\geq 9$ for cohort study were considered as high quality respectively.

## Data processing and statistical analysis

The required data were extracted using Microsoft Excel and imported into STATA version 14 statistical software for meta-analysis. DerSimonian and Laird's random-effects model was used to estimate the overall pooled prevalence of treatment failure [30]. The p-values of

Cochrane Q-test and I2-statics were computed to assess heterogeneity among reported prevalence [31]. According to the results of statistical test, there was significant heterogeneity among the included original studies ($I^2$ = 94.7%, p<0.001), hereafter, a random-effects meta-analysis model was performed to estimate the pooled prevalence. Subgroup analysis was conducted to adjust random variation between point estimates of original study and investigate how failure fluctuates across subgroup participants. Outlier within the included articles was checked using sensitivity analysis. Publication bias across studies was assessed using funnel plot and egger's regression test. The Begg's regression test results at 5% significance level were not statistically significant for publication bias [32]. Forest plot format was used to present the point prevalence and 95% CIs. In this plot, the weight of study indicated by the size of each box, while each crossed line referred to 95% confidence interval. For the secondary outcomes, a log odds ratio was used to determine the association between TF and associated factors.

## Results

### Study selection

We retrieved 291 relevant articles from PubMed, Google Scholar, Science direct, Scopus, and other sources. Of these initial articles, there were about 236 non-duplicated articles. From the remaining articles, 204 articles were excluded after review of their titles and abstracts. Then, 32 potentially full text articles were assessed for eligibility based on the pre-set criteria and 22 articles were further excluded due to different reasons. Lastly, 10 articles met the eligibility criteria and included in the final meta-analysis to determine the prevalence and associated factors of treatment failure (Fig 1).

### Study characteristics

As described in Table 1, the 10 included studies were both cohort and cross-sectional study design, and done from 2003 to 2021. Among the included studies, five studies were conducted in the Amhara region [18,19,24–26], whereas two in Oromia [20,23], two in Addis Ababa [22,27], and one in SNNP [21]. In this meta-analysis study, 4,572 participants were involved to determine the pooled prevalence of treatment failure. The sample size of the studies fluctuated between 96 [20] and 1,186 [22]. Regarding prevalence, the lowest prevalence (3.1%) of treatment failure was reported from a study done in Southern Nations, Nationalities, and Peoples Region (SNNPR) [21], whereas the highest prevalence (18.8%) was reported in study done at Fiche and Kuyu hospitals in Oromia Region [23]. Concerning quality of the included studies, two studies [19,25] were assessed based on JBI check list for cross-sectional studies, and the remaining eight studies [18,20–24,26,27] based on JBI check list for cohort studies. According to this quality assessment criterion, all studies were included.

### Risk of bias in studies

Publication bias was checked using the Begg's test which showed no statistically significant publication bias with p-value of 0.93. We also performed publication bias assessment for overall treatment failure using funnel plot (Fig 2).

### Results of individual studies

**Prevalence of HIV/AIDS treatment failure.** The pooled prevalence of treatment failure among HIV-positive children in Ethiopia was found to be 12.34 (95%CI: 8.59, 16.10) (Fig 3).

Based on clinical definition the pooled prevalence of treatment failure was 27.65 (95% CI: 1.23, 54.06) (Fig 4).

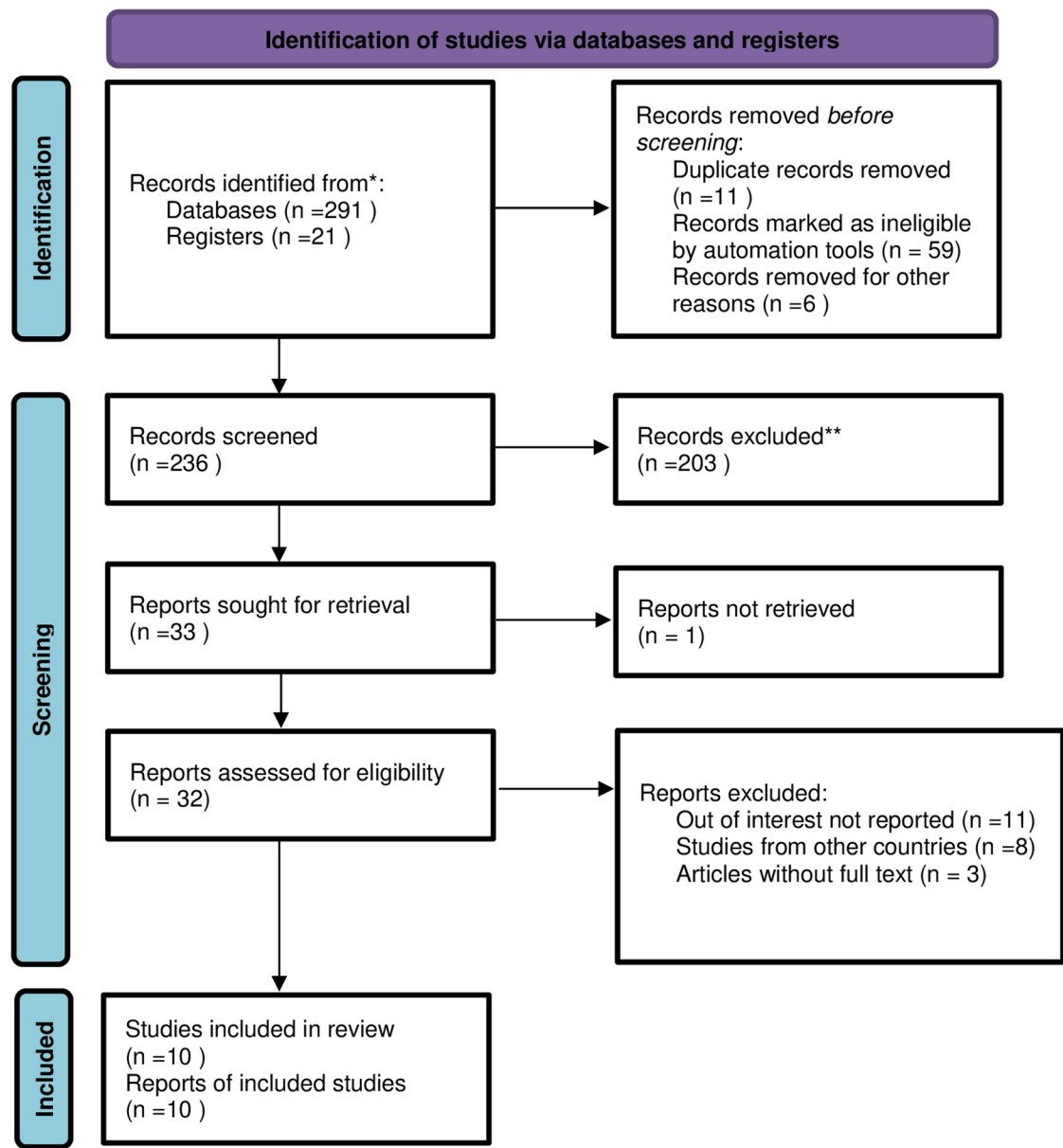

**Fig 1. PRISMA 2020 flow diagram of primary study selection for systematic review and meta-analysis of treatment failure and associated factors among HIV-positive children in Ethiopia.**

Concerning immunological failure, the pooled prevalence was 17.97 (95% CI: 10.13, 25.82) (Fig 5).

Furthermore, the pooled prevalence of virological treatment failure was 23.93 (95% CI: 3.21, 44.65) (Fig 6).

## Heterogeneity and sensitivity analysis

Heterogeneity test ($I^2$) was 94.7%, p < 0.001 which shows that there is significant variation across the included original studies. In the sensitivity analysis, there is no study away from the lower and upper limit of confidence interval.

**Table 1. General characteristics of studies included in systematic review and meta-analysis of treatment failure and associated factors among children on ART in Ethiopia 2021.**

| Author (year) | Study Design | Follow-up (years) | Region | Study area | Sample size | Response rate (%) | Prevalence at 95% CI |
|---|---|---|---|---|---|---|---|
| Zeleke et al (2016) [18] | Cohort | 2005–2013 | Amhara | Gondar | 225 | 100 | 18.2 |
| Bacha et al (2012) [22] | Cohort | 2005–2011 | Addis Ababa | Addis Ababa | 1,186 | 100 | 14.1 |
| Sisay et al (2018) [24] | Cohort | 2010–2016 | Amhara | Amhara | 824 | 81.9 | 7.7 |
| Yassin et al (2017) [23] | Cohort | 2006–2015 | Oromia | Fiche and Kuyu | 269 | 86.8 | 18.8 |
| Netsanet et al (2009) | Cohort | 2005–2008 | Oromia | Jimma | 96 | 100 | 11.5 |
| Tadesse et al (2017) [21] | Cohort | 2015–2016 | SNNP | | 628 | 100 | 3.1 |
| Gelaw et al (2021) [25] | Cross-sectional | 2016–2019 | Amhara | Bahir Dar | 424 | 94.1 | 14.8 |
| Yihun et al (2019) [26] | Cohort | 2011–2018 | Amhara | Debre Markos and Bahir Dar | 402 | 89.8 | 12.19 |
| Haile et al (2019) [27] | Cohort | 2003–2018 | Addis Ababa | Addis Ababa | 391 | 81.3 | 22.6 |
| Getawa et al (2021) [19] | Cross-sectional | 2005–2017 | Amhara | Gondar | 200 | 100 | 12.5 |

## Subgroup analysis

To adjust the reported heterogeneity of the study, subgroup analysis was employed based on regions, study design, and sample size. Higher prevalence of treatment failure was observed in Addis Ababa with a prevalence of 15.92 (95% CI: 11.72, 20.12) followed by Oromia regions at 14.47 (95% CI: 9.79, 19.14) (Fig 7).

On the other hand, the prevalence of treatment failure was lower in studies having a sample of size 628, 3.03 (95%CI: 1.69, 4.37) compared to those having a sample size 391, 18.41 (95% CI: 14.57, 22.26) (Fig 8).

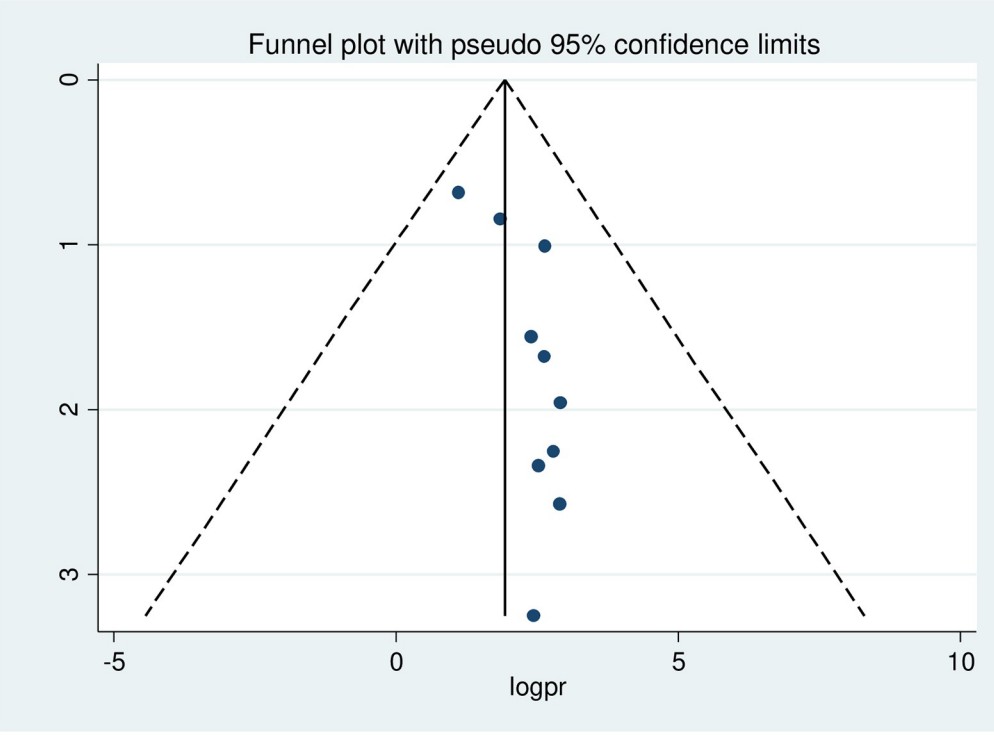

**Fig 2. Funnel plot with 95% confidence limits of the pooled prevalence of treatment failure among HIV positive children in Ethiopia.**

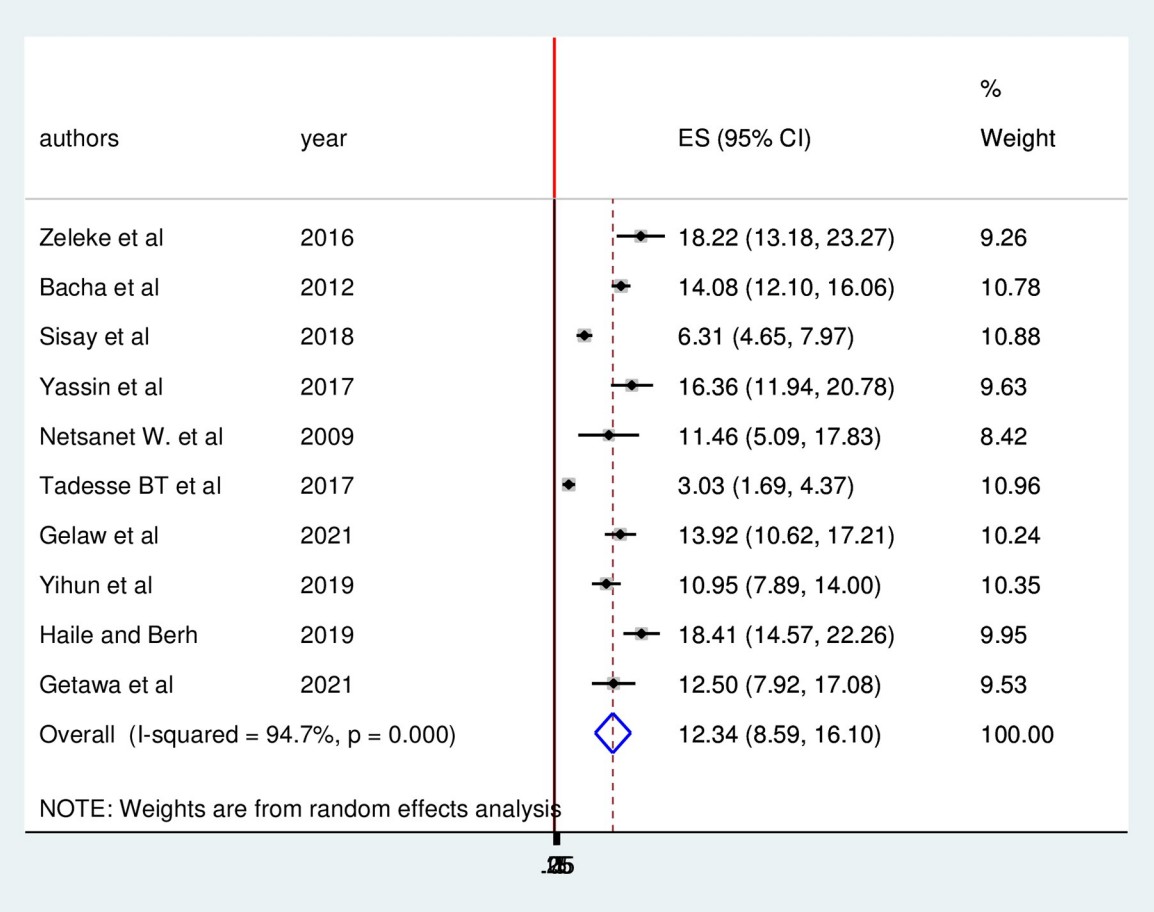

**Fig 3. Forest plot of the pooled prevalence of over-all treatment failure among HIV positive children in Ethiopia.**

Furthermore, we performed subgroup analysis based on study design. Accordingly, the prevalence of treatment failure in cross-sectional study was 13.43 (95% CI: 10.76, 16.11) and in cohort study 12.14(95% CI: 7.83, 16.44) (Fig 9).

### Associated factors of HIV/AIDS treatment failure

In this meta-analysis, the associated factors were categorized in to two thematic areas. These were: 1. Socio-demographic 2. Clinical and drug related factors.

**Socio-demographic factors.** Based on the finding of a study [19], male children were more likely (AOR = 3.15, 95% CI: 1.18, 8.39) to develop HIV treatment failure. From a single study, HIV positive children have not both parents as primary caretakers (AOR = 2.72, 95% CI: 1.05, 7.06], and negative serologic status of their caretakers (AOR = 2.69, 95% CI: 1.03, 7.03) were at high risk of treatment failure [27]. Age of children below 5 years (AOR = 2.4, 95% CI: 1.0–5.7) [25] was also reported as a contributing factor for the occurrence of HIV treatment failure as compared to their counterparts. Likewise, HIV positive children with age below 3 years were at high risk (AHR = 1.85, 95% CI: 1.24, 2.76) to develop treatment failure as compared to those with age between 5 and 15 years [22]. On the other hand, the age of HIV

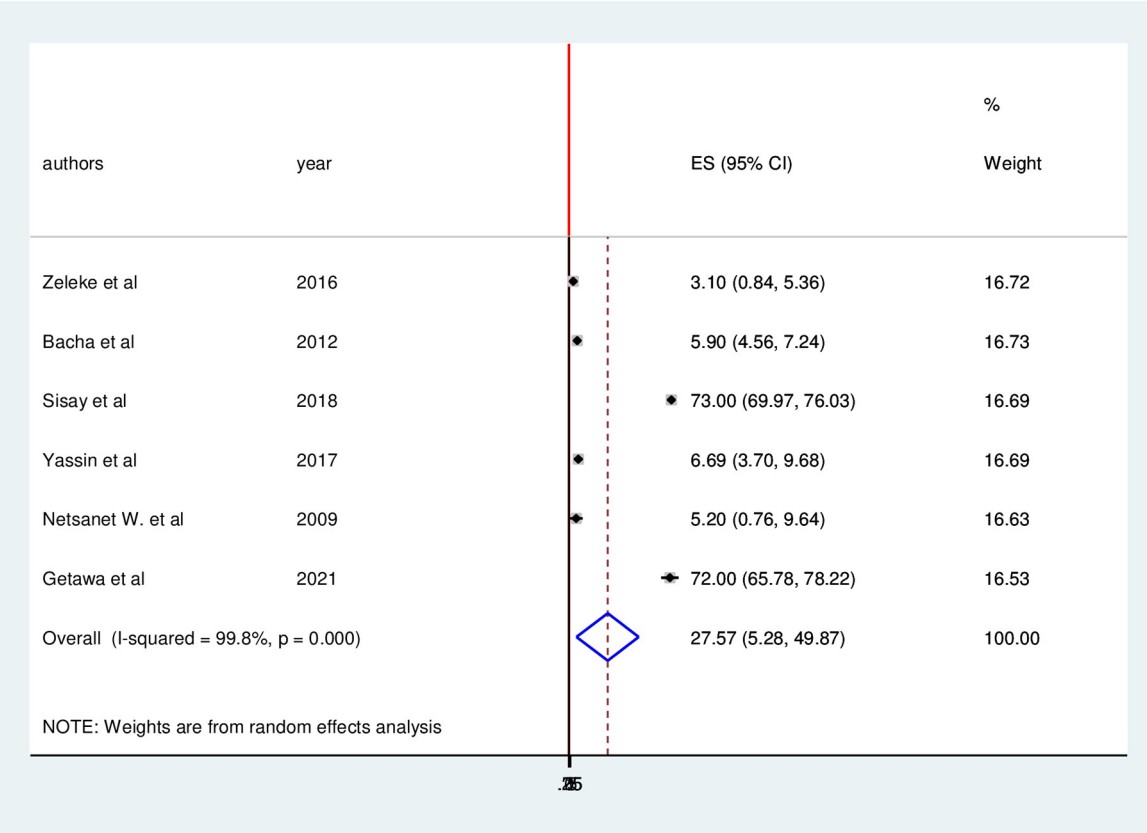

**Fig 4. Forest plot of the pooled prevalence with corresponding 95% CIs of clinical failure among HIV positive children in Ethiopia.**

positive children between 6 and 9 years was protective (AOR = 0.26; 95% CI: 0.09, 0.72) from the occurrence of treatment failure compared to age between 10 and 15 years [23].

**Clinical and drug related factors.** The pooled odds ratio of this meta-analysis revealed that HIV positive children with history of opportunistic infections were 2.6 times more likely (AOR = 2.64; 95% CI: 2.19, 4.31) to develop treatment failure as compared to their counterparts. The estimated pooled effects of advanced WHO clinical stage III/IV on HIV treatment failure was (AOR = 1.66; 95% CI: 1.24, 3.21) as compared to those children categorized on WHO clinical stage I/II. Report from a single study showed that children who interrupt and restart their HIV treatment were positively (AOR = 2.21, 95% CI: 1.09, 4.54) associated with treatment failure [25]. HIV positive children who had height for age below or on 3[rd] percentile were at high risk (AHR = 3.3, 95% CI: 1.0, 10.6) of treatment failure [22]. Additionally, children who was not disclosed their HIV status were also at high risk (AHR = 4.4, 95% CI: 1.8, 11.3) to develop treatment failure [24]. Original regimen change (AOR = 9.22, 95% CI: 3.36, 25.03) [19] and ART drug substitution (AHR = 1.7, 95% CI: 1.1, 2.7) [22] were reported significantly associated factors HIV treatment failure. The result of this meta-analysis revealed that adherence level was positively associated with HIV treatment failure. From this finding, the estimated pooled effect of poor ART adherence level on childhood HIV treatment failure was (AOR = 2.53; 95% CI: 2.03, 4.97) as compared to good adherence level counterparts.

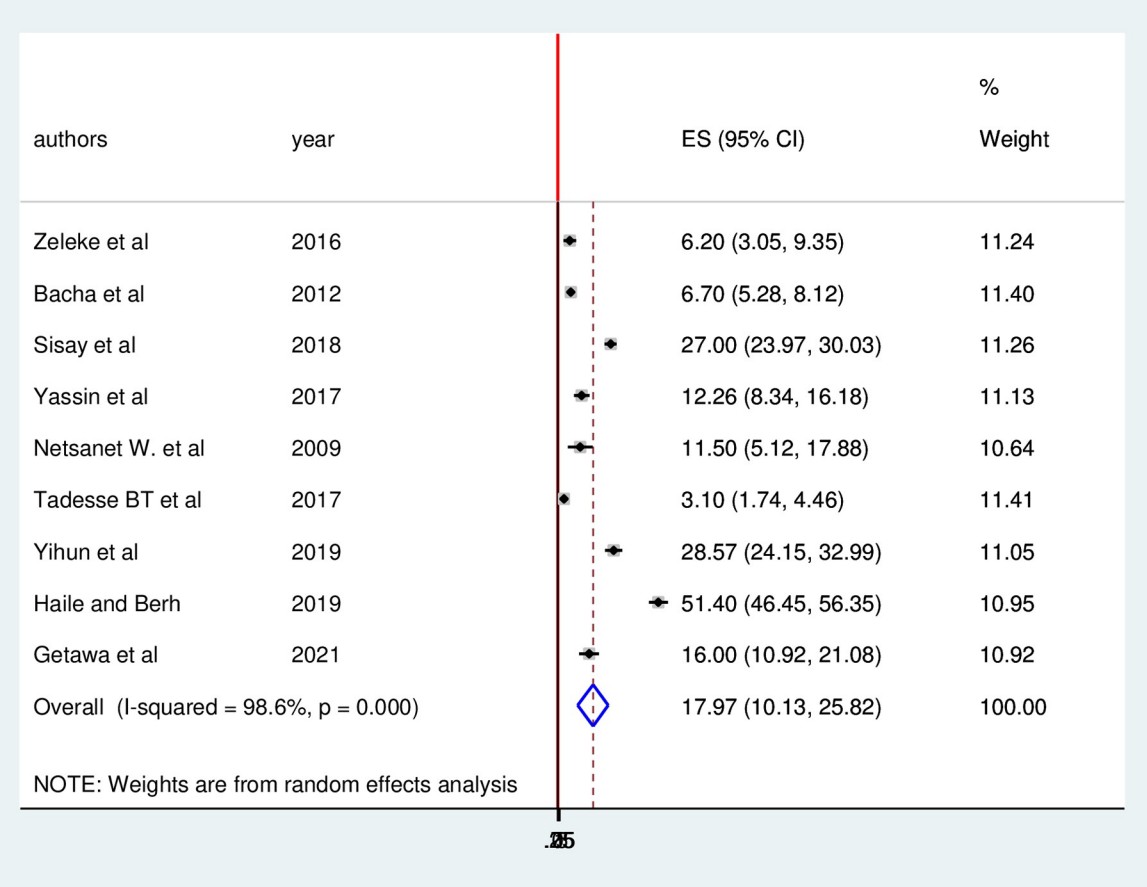

**Fig 5. Forest plot of the pooled prevalence with corresponding 95% CIs of immunological failure among HIV positive children in Ethiopia.**

## Discussion

This study was carried out to estimate the nationwide prevalence and associated factors of treatment failure among children in Ethiopia. In this meta-analysis, the overall pooled prevalence of HIV treatment failure was 12.34 (95%CI: 8.59, 16.10). Globally, HIV treatment failure leads the patient to start the more expensive and toxic ART regimen, develop drug resistant viral strains [33]. It is one of the major causes of childhood morbidity and mortality in developing countries including Ethiopia. To the best our knowledge, this meta-analysis is the first of its kind to estimate the pooled prevalence of treatment failure and its associated factors among HIV positive children in Ethiopia. Hence, the result of this review will help the country to keep children on first line treatment regimen.

The finding is in agreement with studies conducted in Malawi (16%) [34] and Ghana (15.7%%) [35]. However, this finding is much lower than studies conducted in Tanzania (25.4%) [36], Uganda and Mozambique (29%) [37], and Kenya (43.1%) [38]. On the other hand, our finding is much higher than a study done in Ghana (6.5%) [39]. The possible explanation for the observed variations might be due to differences in methodology and sample size used to diagnose treatment failure by individual studies conducted in each country. Moreover,

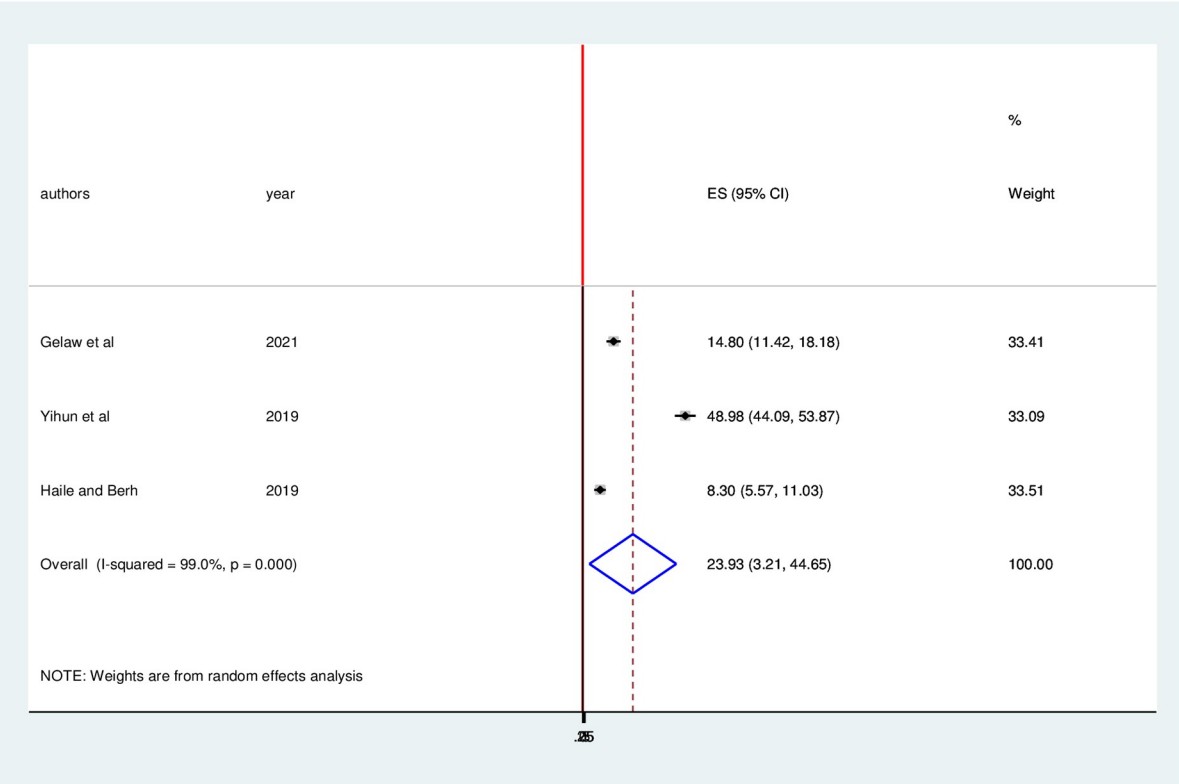

**Fig 6. Forest plot of the pooled prevalence with corresponding 95% CIs of virological failure among HIV positive children in Ethiopia.**

the difference could be due to the difference in geographical areas, quality of medical service, and socio-economic status which have an unlimited effect to diagnose HIV treatment failure among children.

There was statistically significant heterogeneity across the original studies included in this review. Therefore, we did subgroup analysis. Accordingly, the subgroup analysis of this study revealed that the highest prevalence of HIV treatment failure was observed in Addis Ababa, 15.92 (95% CI: 11.72, 20.12) followed by Oromia region, 14.47 (95% CI: 9.79, 19.14) whereas the lowest prevalence was identified in SNNPR with a prevalence of 3.03 (95%CI: 1.69, 4.37). The possible explanations for this variation might be due to the difference in sample size, study design, and number of studies included in this review from each region.

In this review, we also investigated factors associated with HIV treatment failure among children taking ART in Ethiopian. Poor ART adherence level, classified as advanced WHO clinical stage III/IV, and history of opportunistic infections were found be the leading factors for the occurrence of childhood treatment failure.

HIV positive children with poor ART adherence level were more likely to experienced treatment failure. After ART is started to HIV infected individuals, it should take for life long without any interruption. If the drug is not taken daily, on time, and regularly; patients are at higher risk drug resistance viral strain. It is widely agreed that good ART adherence has positive impact for suppression of viral replication and improve clinical as well as immunological outcomes which in turn helps to minimize the risk of HIV drug resistance [40]. Approximately

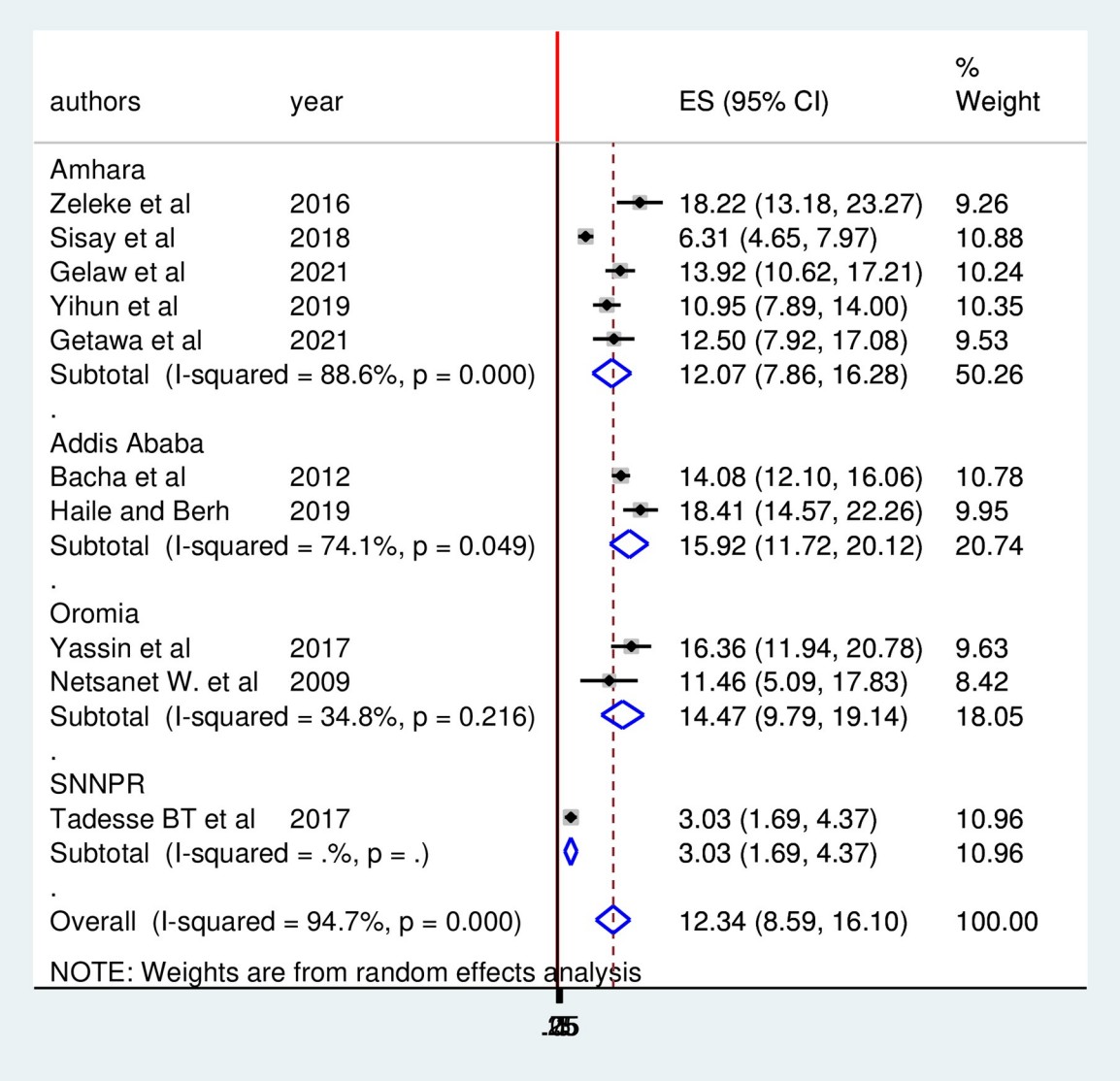

**Fig 7. Forest plot of the prevalence of treatment failure with corresponding 95% CIs of the subgroup analysis based on the regions, where the studies done.**

11.3% of HIV positive children have poor ART adhere level which is continued to be a devastating clinical challenge in Ethiopia [41].

Advanced WHO clinical stage is another determinant factor for HIV treatment failure. HIV positive children classified as WHO clinical stage III/IV are more likely to develop treatment failure. Children interrupt to take their medication and lost to follow up which makes them highly vulnerable for opportunistic infections and advanced WHO clinical stage. This makes them to be exposed for HIV treatment failure.

Furthermore, children with lower CD4 cell count and history of opportunistic infections were found to be at high risk to develop HIV treatment failure. This is due to the fact that there is linkage between opportunistic infections and level CD4 cell counts. In human being

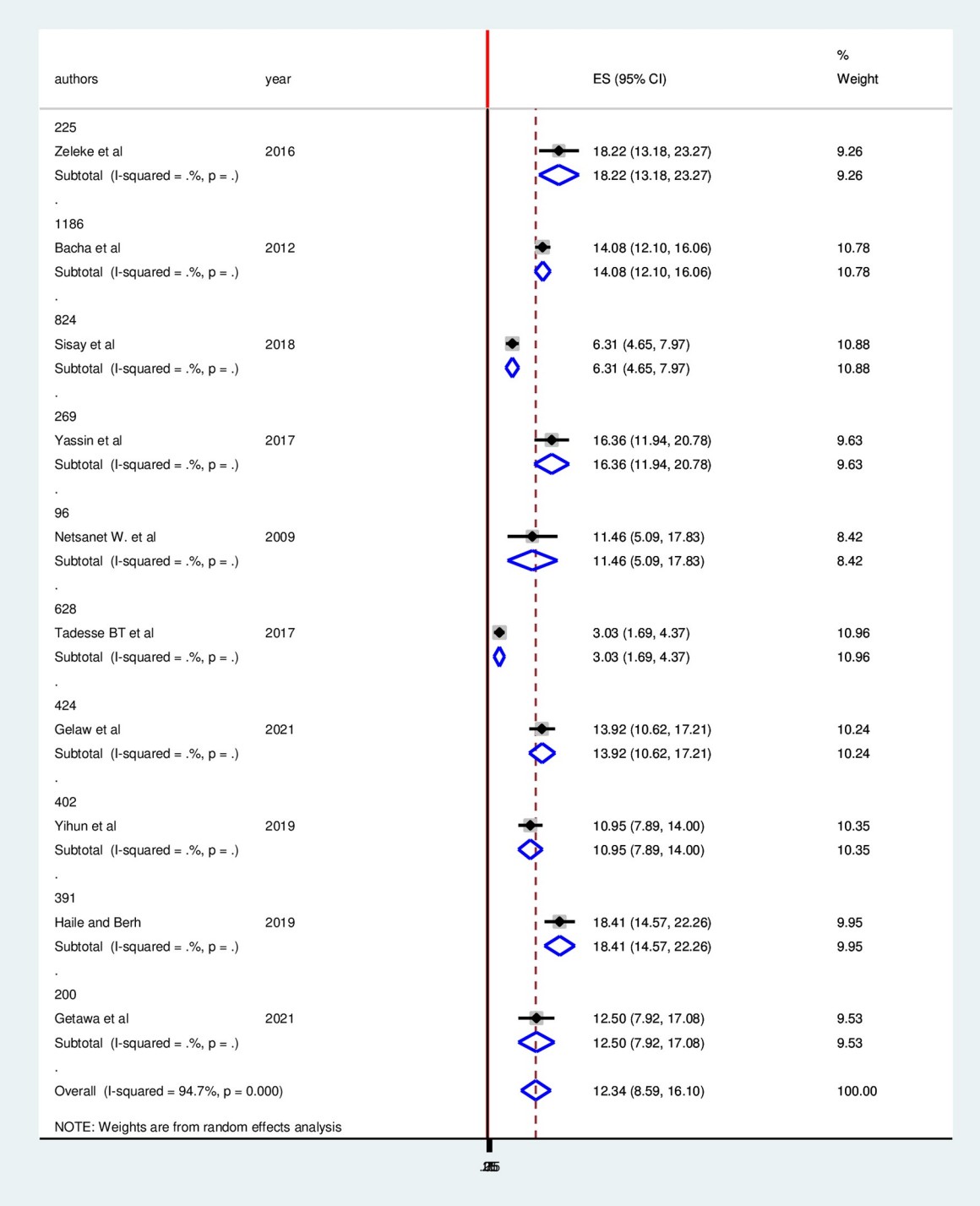

**Fig 8. Forest plot of the prevalence of treatment failure with corresponding 95% CIs of the subgroup analysis based on sample size.**

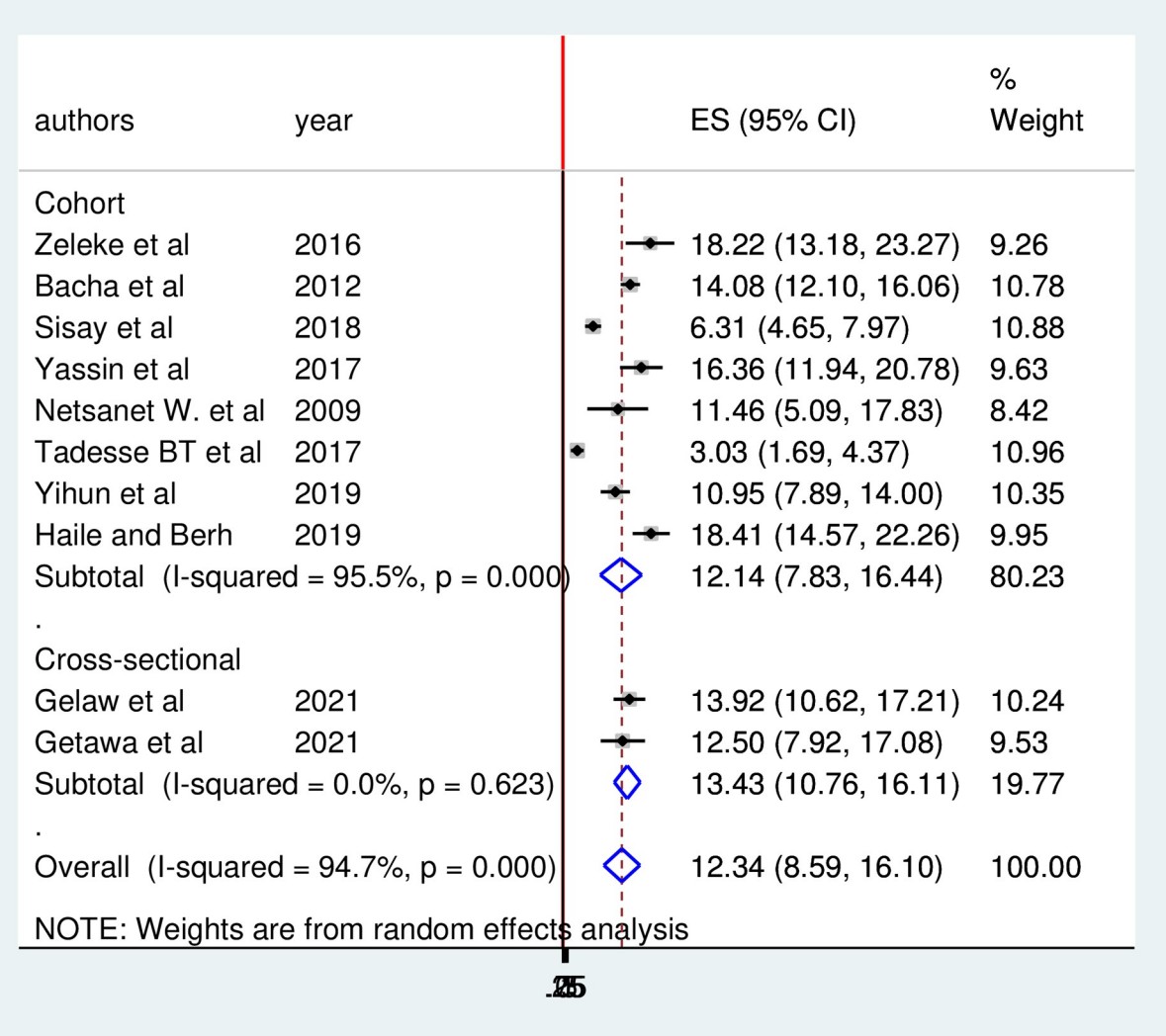

**Fig 9. Forest plot of the prevalence of treatment failure with corresponding 95% CIs of the subgroup analysis based on the study design.**

CD4 cells have its own role on the production of immunity which protects the body from disease causing pathogens. As CD4 cell count decrease, the rate of viral replication on the other hand becomes increased [42,43]. In addition, the care givers as well as patients focus on the current co-infection, missed/interrupt to take their ART drug and follow up which results to develop treatment failure.

## Limitations of the review

This review was included only articles studied in the English language, which may limit some papers from being included. Besides, this study represented only studies conducted from three regions and one administrative town of the country, which could under-represent the HIV treatment failure proportion for the entire nationwide context. Moreover, this meta-analysis includes limited number of studies; the result may not represent the nationwide figure of failures.

## Conclusion

In this meta-analysis, treatment failure among HIV positive children in Ethiopia was found to be significantly high. Poor ART adherence level, opportunistic infections, and advanced WHO clinical stage were factors significantly associates with treatment failure among HIV positive children in Ethiopia. Thus, based on our results, we recommend particular emphasis shall be given to prevent opportunistic infection and improve treatment adherence level which in turn helps to tackle HIV drug resistance and keep on first line regimen. The result of this review may have implication on clinical and health policy system. Preventing opportunistic infections and improving ART adherence level will be used to increase treatment outcome. Additionally, the finding will help to monitor the national program towards the success of UNAIDS global target of 90-90-90.

## Supporting information

**S1 Checklist. PRISMA 2020 checklist.**
(DOCX)

**S1 File.**
(XLSX)

## Acknowledgments

We would like to give our special gratitude for the authors of primary studies which were incorporated to this systematic review and meta-analysis.

## Author Contributions

**Conceptualization:** Belete Gelaw, Amare Kassaw.

**Data curation:** Belete Gelaw, Tiwabwork Tekalign, Bogale Chekole.

**Formal analysis:** Belete Gelaw, Tiwabwork Tekalign.

**Methodology:** Belete Gelaw, Lemma Dessalegn, Tadele Lankirew, Kirubel Eshetu, Chalie Marew.

**Project administration:** Bogale Chekole, Amare Kassaw.

**Software:** Belete Gelaw, Eyasu Alem.

**Supervision:** Kirubel Eshetu.

**Validation:** Belete Gelaw.

**Visualization:** Bogale Chekole.

**Writing – original draft:** Belete Gelaw.

**Writing – review & editing:** Belete Gelaw, Lemma Dessalegn, Tiwabwork Tekalign.

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
