## [Decision Letter · Decision Letter 0]

18 Nov 2021

PONE-D-21-32599Prevalence and associated factors of treatment failure among children on ART in Ethiopia: a systematic review and meta-analysisPLOS ONE

Dear Dr. Gelaw,

Thank you for submitting your manuscript to PLOS ONE. After careful consideration, we feel that it has merit but does not fully meet PLOS ONE’s publication criteria as it currently stands. Therefore, we invite you to submit a revised version of the manuscript that addresses the points raised during the review process.

There are major comments from the reviewer, which includes the need for additional analyses.

We look forward to receiving your revised manuscript.

Kind regards,

Frank T. Spradley

Academic Editor

PLOS ONE

Journal Requirements:

"No funding was obtained for this review"

Reviewers' comments:

Reviewer's Responses to Questions

**Comments to the Author**

1. Is the manuscript technically sound, and do the data support the conclusions?

Reviewer #1: Yes

2. Has the statistical analysis been performed appropriately and rigorously? 

Reviewer #1: Yes

3. Have the authors made all data underlying the findings in their manuscript fully available?

Reviewer #1: Yes

4. Is the manuscript presented in an intelligible fashion and written in standard English?

Reviewer #1: Yes

5. Review Comments to the Author

Reviewer #1: Dear Authors,

You have presented useful data.

I have the following comments

It will be worthwhile to do a sub-group/sensitivity analysis of cohort and cross-sectional analysis. As, in you may exclude cross-sectional studies and check for the estimates in cohort studies

Repeat it for cross-sectional and compare whether there is a difference

You may also look at Urban/Rural population

Any variables for meta-regression?

6. PLOS authors have the option to publish the peer review history of their article (what does this mean?). If published, this will include your full peer review and any attached files.

Reviewer #1: No

---

## [Author Response · Author response to Decision Letter 0]

2 Dec 2021

Reviewer #1 

Comments to the Author

The authors of this systematic review and meta-analysis have presented useful data to determine the pooled national burden of childhood HIV/AIDS treatment failure as well as its associated factors in Ethiopia. Conversely, there are specific comments that the reviewer would like the authors to address so as to further improvement of the manuscript.

Authors’ response: we are very glad to the reviewer’s appreciation of our efforts; and we have just given our respective responses to each of the specific reviewer comments as detailed below.

1. Is the manuscript technically sound, and do the data support the conclusions?

Thank you very much for your insightful comment. Yes. We have a conclusion which is supported by the data analysed and presented in this study (page 16, line 327-336). 

2. Has the statistical analysis been performed appropriately and rigorously?

We acknowledge the reviewer’s concern regarding to statistical analysis for this study. Yes. We have carried out all the necessary statistical analysis properly and presented our finding accordingly in the main documents of this study.

3. Have the authors made all data underlying the findings in their manuscript fully available?

Yes. Comments have been taken and we have made the necessary correction accordingly (page 18, line 357-358). For detail please see the supporting information file.

4. Is the manuscript presented in an intelligible fashion and written in standard English?

Great thanks. We have carefully reviewed the manuscript based on your suggestion and some typographic error has been also updated (page 7, line 153).

5. It will be worthwhile to do a subgroup/sensitivity analysis of cohort and cross-sectional analysis. As, in you may exclude cross-sectional studies and check for the estimates in cohort studies. Repeat it for cross-sectional and compare whether there is a difference. You may also look at Urban/rural population. Any variable for meta-regression?

Absolutely! We have taken the given comment and sensitivity analysis was already done accordingly. However, the result indicated that there is no single study unduly influenced the overall estimates of treatment failure and there is no significant variability for meta-regression. We have done subgroup analysis based on study design, and giving a prevalence of 12.14(95% CI: 7.83, 16.44) in cohort which is nearly the same in cross-sectional study 13.43 (95% CI: 10.76, 16.11) (page 12, line 226-237). 

NB. Generally, we have tried to revise our manuscript according to the requirement of PLOS ONE template.

---

## [Editor Report · Decision Letter 1]

7 Dec 2021

Prevalence and associated factors of treatment failure among children on ART in Ethiopia: a systematic review and meta-analysis

PONE-D-21-32599R1

Dear Dr. Gelaw,

We’re pleased to inform you that your manuscript has been judged scientifically suitable for publication and will be formally accepted for publication once it meets all outstanding technical requirements.

Kind regards,

Frank T. Spradley

Academic Editor

PLOS ONE

---

## [Editor Report · Acceptance letter]

10 Jan 2022

PONE-D-21-32599R1 

Prevalence and associated factors of treatment failure among children on ART in Ethiopia: a systematic review and meta-analysis 

Dear Dr. Gelaw:

I'm pleased to inform you that your manuscript has been deemed suitable for publication in PLOS ONE. Congratulations! Your manuscript is now with our production department. 

Kind regards, 

on behalf of

Dr. Frank T. Spradley 

Academic Editor

PLOS ONE